# Identification of Genomic Regions Implicated in Susceptibility to *Schistosoma mansoni* Infection in a Murine Backcross Genetic Model

**DOI:** 10.3390/ijms241914768

**Published:** 2023-09-30

**Authors:** Juan Hernández-Goenaga, Julio López-Abán, Adrián Blanco-Gómez, Belén Vicente, Francisco Javier Burguillo, Jesús Pérez-Losada, Antonio Muro

**Affiliations:** 1Grupo de Enfermedades Infecciosas y Tropicales (e-INTRO), IBSAL-CIETUS (Instituto de Investigación Biomédica de Salamanca, Centro de Investigación de Enfermedades Tropica-les de la Universidad de Salamanca), Facultad de Farmacia, Universidad de Salamanca, Ldo. Méndez Nieto s/n, 37007 Salamanca, Spain; juhego@usal.es (J.H.-G.); belvi25@usal.es (B.V.); 2Instituto de Investigación Biomédica de Salamanca (IBSAL), Complejo Asistencial Universitario de Salamanca, Hospital Virgen de la Vega, 37007 Salamanca, Spain; adrianblanco@usal.es (A.B.-G.); jperezlosada@usal.es (J.P.-L.); 3Instituto de Biología Molecular del Cáncer (IBMCC), Centro de Investigación del Cáncer (CIC)—CSIC, Laboratory 20, 37007 Salamanca, Spain; 4Departamento de Química-Física, Facultad de Farmacia, Universidad de Salamanca, C/Donantes de Sangre s/n. Campus Unamuno, 37007 Salamanca, Spain

**Keywords:** *Schistosoma mansoni*, backcross schistosomiasis, schistosomiasis, pathology, genetic susceptibility, genomic region

## Abstract

Only a small number of infected people are highly susceptible to schistosomiasis, showing high levels of infection or severe liver fibrosis. The susceptibility to schistosome infection is influenced by genetic background. To assess the genetic basis of susceptibility and identify the chromosomal regions involved, a backcross strategy was employed to generate high variation in schistosomiasis susceptibility. This strategy involved crossing the resistant C57BL/6J mouse strain with the susceptible CBA/2J strain. The resulting F1 females (C57BL/6J × CBA/2J) were then backcrossed with CBA/2J males to generate the backcross (BX) cohort. The BX mice exhibited a range of phenotypes, with disease severity varying from mild to severe disease, lacking a fully resistant group. We observed four levels of infection intensity using cluster and principal component analyses and K-means based on parasitological, pathological, and immunological trait measurements. The mice were genotyped with 961 informative SNPs, leading to the identification of 19 new quantitative trait loci (QTL) associated with parasite burden, liver lesions, white blood cell populations, and antibody responses. Two QTLs located on chromosomes 15 and 18 were linked to the number of granulomas, liver lesions, and IgM levels. The corresponding syntenic human regions are located in chromosomes 8 and 18. None of the significant QTLs had been reported previously.

## 1. Introduction

Schistosomiasis is a widespread parasitic disease caused by trematodes of the genus *Schistosoma* in tropical areas. It is estimated that more than 200 million people worldwide are infected by *Schistosoma* spp., which caused over 1.4 million DALYs (disability-adjusted life years) in 2017 [1]. Schistosomiasis is a complex disease resulting from the parasite’s interaction with endogenous host factors such as the intensity and duration of infection, nutritional status, and other associated diseases [2]. Significant heterogeneity has been observed in everyone’s response to the parasite in the endemic areas. Only a minority of patients suffer severe hepatic fibrosis or have a high infection burden [3]. Based on studies of single-nucleotide polymorphisms (SNPs) and microsatellite marker studies, a specific chromosomal region known as SM1 (*S. mansoni* 1) locus has been identified in endemic regions of Brazil and Senegal. This locus is associated with higher prevalence and greater worm burden, particularly within certain family groups [4,5]. The genetic region associated with susceptibility, the SM1 locus, is localized in the long arm of human chromosome 5 (5q31-33). Another locus associated with severe schistosomiasis, referred to as SM2 (*S. mansoni* 2), is linked to the development of liver fibrosis and is located on the long arm of human chromosome 6 (6q22-23) [6,7]. However, there have been reports suggesting genomic regions linked to high infection intensity and severe liver damage. There is no comprehensive information regarding the specific genomic regions involved in these processes. Knowledge of molecular markers associated with susceptibility could be helpful in developing protocols for the diagnosis and treatment of severe schistosomiasis. Additionally, it would enable the design of effective control measures aimed at preventing infection in more susceptible populations.

The identification of genetic markers of complex diseases like schistosomiasis is difficult within the human population. It requires population studies involving numerous cases and controls and a great expense of time and resources without guaranteeing success. However, inbred mouse strains offer a valuable alternative as they have genetic and phenotypic uniformity in schistosomiasis susceptibility and other traits [8]. Also, crosses between inbred mouse strains generate cohorts of individuals with a wide range of susceptibility to schistosomiasis This approach allows for the identification of genetic regions, such as quantitative trait loci (QTL) and expression QTLs (eQTLs), that are associated with variations in quantitative traits related to resistance or susceptibility to disease [9]. Identifying these genetic loci linked to the disease susceptibility will help elucidate the molecular basis of this complex disease [10,11]. Indeed, linkage studies with SNPs have identified chromosomal regions associated with the genetic influence of schistosomiasis and its wide variability in severity. It has been possible to delimit genetic regions involved both in the intensity of the schistosome infection and in the degree of liver fibrosis induced in an individual [12,13,14]. In laboratory mice, *S. mansoni* infections lead to granulomatous lesions in the intestine and liver. We carried out a preliminary study with five inbred mouse strains (C57BL/6J, FVB/NH, DBA/2J, BALB/c, and CBA/2J), and we identified the two most divergent ones in terms of infection susceptibility: CBA/2J exhibited higher susceptibility, while C57BL/6J showed lower. The F1B5CBA hybrid mice from these two parental strains showed intermediate susceptibility to infection [15]. The strain differences in response to *S. mansoni* among different mouse strains offer the chance to study the genetic factors that influence whether an animal will develop intense or mild worm recovery or lesions to this infection. The evidence of differences between more and less susceptible mouse strains led us to adopt a backcrossing strategy of C57BL/6J onto the CBA/2J background. This approach should generate a mouse cohort with high phenotypic variability between the two strains.

The objective of this work was to identify chromosomal regions that carry low penetrance or modifiers genes associated with the susceptibility to *S. mansoni* infection using a murine backcross genetic model. Then, we characterized the susceptibility to *S. mansoni* infection by defining different disease patterns; studied the associations between the pathophenotypes and the related parasitological, pathological, and immunological traits; and identified genomic regions (QTLs) linked to the intermediate phenotypes observed monitoring *S. mansoni* infection.

## 2. Results

### 2.1. There Was a Different Susceptibility to Schistosomiasis in Mice Related to the Genetic Background but Not to Mouse Sex

In a previous study, we assessed the susceptibility–resistance to *Schistosoma mansoni* infection in C57BL/6J and CBA/2J mouse strains, and the F1B6CBA hybrid strain [15]. In this current study, we generated a backcross (BX) cohort comprising 105 mice to identify the chromosomal regions associated with *S. mansoni* infection. The BX cohort presented no significant male worm and female worm counts after perfusion compared with those of the F1B6CBA and had similar numbers to the resistant parental strain. Regarding the number of eggs per gram of liver, the mean was like the F1B6CBA cohort and intermediate to that of the parental strains. Additionally, the number of granulomas in the BX cohort was slightly higher than that in the F1B6CBA hybrid and similar to the susceptible CBA/2J parental strain (Appendix A).

Upon comparing BX males and female mice, we found no significant differences in the number of male worms recovered, eggs trapped in tissues, fecundity, number of granulomas, or injured liver surface (Appendix A). Moreover, boxplots revealed a wide variability in the data collected from different BX mice, reflecting the range between resistant C57BL/6J and susceptible CBA/2J strains (Appendix A).

### 2.2. Parasitological, Pathological, and White Blood Cell Populations Correlated in the BX Cohort

Most of the parasitological and pathological variables showed a direct association between them. However, the number of recovered male and female worms, as well as worm fecundity, showed an inverse association. The hepatic damage variables such as granulomas and affected liver surface, along with the number of peripheral blood lymphocytes at week nine post-infection and the number of eggs in tissues, showed an inverse correlation with splenocyte subpopulations. Moreover, the parasitological and pathological variables were directly associated with immunoglobulin serum levels. The white blood cell subpopulations showed a direct association among themselves but not with immunoglobulin levels except with B220^+^ splenocytes and IgM, which showed an inverse association. Furthermore, the immunoglobulins showed a positive association among themselves, except IgG2a with IgG nor IgM (Figure 1).

### 2.3. Identification of Different Degrees of Schistosomiasis Disease in the BX Mouse Cohort

Multivariant analysis techniques were used to group the mice and analyze the variables’ influence. Initially, a clustering analysis (CA) was performed to study similarities in the BX cohort using recoveries of male and female worms, eggs in the liver and intestine, fecundity, granulomas, and surface liver damage. The distances between cases, represented as branches in a dendrogram, indicate different groups. The data from the BX cohort ranging from the more to the less susceptible mice revealed four groups. However, there were atypical individuals, and establishing a clear threshold was challenging (Appendix A). Then, we applied principal components analysis (PCA) to condense the information into two dummy variables and determine each variable’s load. Principal component 1 (PC1) captured 42% of the information, while PC2 accounted for 28%. Then, we shorted our BX cohort into four groups in agreement with the results obtained using cluster analysis (Appendix A). Moreover, we observed that all variables influenced PC1 in the same sense. On the other hand, the loads of male and female counts had opposing effects on fecundities in PC2. The variables of eggs in tissues and granulomas had a neutral influence on PC2 (Appendix A).

In clusters using the k-means algorithm, initial mean values had to be assigned for each of the four groups. Therefore, we used the mean values obtained from PCA as the starting points. Through iterations, the algorithm joined the cases around the centroids of each group. In Figure 2, centroids and cases are connected through radii. The mice were arranged into each group based on the severity of the infection, ranging from mild (Group 1) to severe (Group 4).

### 2.4. Pathophenotypic Variation in the BX Cohort in the Groups Identified

We evaluated the distribution of schistosomiasis pathophenotypes and intermediate phenotypes among the four identified groups in the BX cohort using multivariate analysis. We considered male and female recoveries, eggs in the liver and intestine, granulomas, and affected liver surface (Figure 3). Furthermore, we studied lymphocyte subpopulations in splenocytes and peripheral lymphocytes (CD4, CD8, B220, and CD45), and immunoglobulin responses (IgG, IgG1, IgG2a, and IGM) (Appendix A). We also analyzed medians in a heatmap, with the maximum median of each variable within the four groups as a reference (Figure 4).

The data showed that Group 1 comprised mice with the lowest worm recovery, eggs in tissues, and liver damage, representing a less severe disease (Figure 3A–H), as supported by the lowest medians (Figure 4). In Group 2, parasitological and pathological variables were significantly higher than in Group 1 (Figure 3A–D), but the count of eggs in tissues was similar to Group 1 (Figure 3E–H). Therefore, Group 2 included mice with a more aggressive disease. Group 3 showed a higher schistosome worm burden than Groups 1 and 2 (Figure 3A,B) and significant liver damage, eggs in the liver, or eggs in the small intestine compared with Group 1 (Figure 3C–F), pointing to a phenotype with more intense disease than Groups 1 and 2, with the highest medians similar to Group 4 (Figure 4). Finally, Group 4 had the highest liver damage levels, counts in eggs in tissues, and fecundity (Figure 3C–H and Figure 4), indicating the most aggressive disease presentation. Interestingly, however, the number of recovered worms was similar to that of Group 1 (Figure 3A,B).

Regarding cell populations, there were no significant differences among the four studied groups for CD4 and CD8 populations, whether in splenocytes or peripheral blood (Appendix A). Group 4, with the highest liver damage and eggs trapped in tissues, showed the lowest percentages in B220 and CD45 splenocytes (Appendix A). These differences were statistically significant when compared to Groups 1 and 2. Moreover, the median analysis is in concordance (Figure 4). In contrast, peripheral blood B220 and CD45 had no significant differences (Appendix A). Finally, we only found differences in total IgG antibodies where Groups 3 and 4 showed more absorbance than Group 1 (Appendix A), showing the highest medians (Figure 4). On the other hand, no significant differences were detected in IgG1, IgG2a, or IgM levels among the groups.

### 2.5. Identifying Chromosomal Regions Associated with Susceptibility

In the BX backcross cohort, after quality control, three SNPs were excluded from the analysis, and six mice with less than 5 or more than 20 overcrossings were eliminated. This resulted in a final database with 86 individuals, 958 SNPs, and 20 phenotypes. Through our analysis, we identified 19 QTL genomic regions associated with the heterogeneous presentation of the disease (Figure 5). These regions were associated with parasite pathological variables (Appendix A), lymphocyte subpopulations in peripheral blood and spleen (Appendix A), and immunoglobulins (Appendix A). In our study, both the SNP rs6169611 on chromosome 15 and SNP rs13483259 on chromosome 18 showed strong associations with the number of granulomas, which is the most representative variable of liver damage. The region including SNP rs6169611 on mouse chromosome 15 corresponded to the human chromosome 8 (8q24.39 (Appendix A), while the region containing SNP rs13483259 on mouse chromosome 18 corresponded to the human chromosome 18 (18q11.2-12.3).

### 2.6. Identification of Genomic Regions Linked to Disease Susceptibility across the Four Groups Identified in the BX

As the genetic component influences disease progression, we conducted a linkage analysis of the QTLs associated with schistosomiasis in the four defined groups within the BX cohort. We identified 11 loci associated with the disease out of the 19 identified. Group 4 included a significantly higher percentage of mice that were heterozygous for QTL11 located on chromosome 5, QTL5 on distal chromosome 6, and QTL7 on chromosome X. Regarding the QTL3 located on chromosome 2, Group 4 had the lowest percentage of heterozygotes, whereas Group 3 had the highest percentage. We also observed that, for QTL8 on chromosome 8 and QTL4 on chromosome 17, Groups 3 and 4, which exhibited the most severe disease, had the highest percentage of heterozygous mice. Similarly, Group 1 showed the lowest percentage of heterozygotes for QTL4 on chromosome 17. Furthermore, for the distal QTL5 on chromosome 6, Group 3 had the lowest number of heterozygous mice, while Group 4 showed the highest percentage of heterozygotes (Figure 6).

## 3. Discussion

Only a minority of chronically exposed patients develop the most severe clinical form of schistosomiasis, characterized by severe hepatosplenic disease, periportal fibrosis, and portal hypertension. Schistosomiasis is the second most prevalent helminthiasis after soil-transmitted nematodes worldwide, with a high number of infected people [16,17]. Hepatic granulomas and subsequent liver fibrosis vary significantly from individual to individual. Schistosome–host interaction is influenced by multiple factors, such as the type of immune response developed by the host, genetic background, intensity, and the frequency of infections. In endemic areas, high infection levels and severe periportal fibrosis are found to be concentrated in some families. These differences are not attributed to single-gene inheritance of a single dominant or recessive gene but rather to the involvement of multiple genes, indicating the complex nature of these traits [18]. The susceptibility to infectious diseases responds to a model of complex traits, where the challenge lies in the fact that the same genotype can determine diverse phenotypes. This is because complex traits often exhibit incomplete penetrance and are influenced by random factors and environmental conditions [19]. There are publications where experimental *Schistosoma mansoni* infection is studied in different mouse strains [20,21,22]. These studies analyze histopathological factors such as granulomas and hepatic fibrosis, but they do not identify specific genetic markers associated with susceptibility or resistance to infection.

In this study, we examined the schistosomiasis behavior in a genetically heterogeneous population created through backcrossing between a less susceptible (C57BL/6J) and a more susceptible (CBA/2J) mouse strains to generate a genetically and phenotypically heterogeneous BX cohort. Parasitological, pathological, and magnitudes were studied to determine the level of infection, lesions, and immune response [23]. The results obtained from the BX mice showed worm recoveries similar to those observed in the resistant parental strain. Also, eggs in tissues and liver damage were found to be intermediate between the parental strains, consistent with previous studies conducted by our research group [24]. When comparing the phenotypic variability of the BX mice with the F1B6CBA, we observed a lower number of both male and female worms, as well as a decreased number of eggs per gram of liver [15]. After comparing the male and female worms in the BX cohort, we did not observe any significant differences in the studied parasitological or pathological variables. This indicates that there is no association between the pathology caused by *S. mansoni* and the sex of the mice. Overall, the data showed significant variation in the expected range for the BX cohort. Furthermore, a correlation analysis between 20 phenotypic variables was performed. The parasitological and pathological measurements showed a positive correlation among each other, except for the number of worms and fecundity, which showed an inverse relationship. Specifically, a higher number of female worms was associated with less egg production [25]. In the same way, the immunological data showed correlations among themselves, but there was a low correlation with parasitological or pathological variables, indicating an indirect influence of tissue damage on immune cell populations. The immune response to *S. mansoni* infection is very complex [23] since different types of cells of the immune system are involved, orchestrated by linker or effector molecules such as immunoglobulins, which vary depending on the disease stage or biological cycle phase [26].

We could define four groups or clusters of animals with different levels of disease according to the phenotypic variables analyzed, employing cluster analysis dendrograms, principal component analysis, and k-means clusters. As expected, in our model of phenotypic variability, the variables of eggs in tissues and hepatic damage had the most important loads in determining the severity of disease across the four groups, ranging from mild to severe. These findings indicate that adult worms are unrelated to histopathological lesions. Instead, egg laying triggers the inflammatory reaction and granuloma formation. We also observed that the CD4 and CD8 cell subpopulations were predominant in the group with less and intermediate disease severity, both peripheral blood and spleen. However, their presence significantly decreased in the most severe group. CD4 lymphocytes are essential for granuloma formation and orchestrate the immune response against *S. mansoni* [27,28]. On the other hand, CD8 lymphocytes modulate the Th2 immune response [29]. In our study, we observed the production of antibodies against the infection, but we did not observe any differences toward a Th1 or Th2 response associated with disease severity [30,31]. The BX cohort showed a broad range of variability from mild to severe infection intensity and disease. This variation makes it a suitable model for studying schistosomiasis across its gradation. Using this model, we could associate pathophenotypes and intermediate phenotypes with specific chromosomal regions using SNP markers.

The linkage analysis was performed using 958 differential SNPs with 86 mice and 20 variables after quality controls to detect genotyping errors, quantify crossovers, and generate a suitable genetic map [32]. Once the recombination distances in the BX cohort were obtained, the QTL regions linked to the pathophenotypes were identified. Through this analysis, we identified 19 chromosomal regions associated with the different phenotypic studied variables with an LOD score above 1.4. The most noteworthy QTLs were QTL1 and QTL2 located on chromosomes 15 and 18, respectively. QTL1 was associated with three SNPs on chromosome 15 based on granulomas, affected liver surface, and spleen CD8 population. On the other hand, QTL2 was associated with chromosome 18 through a single SNP with a high LOD score linked to granulomas and liver damage. QTL3 and QTL4 were located on chromosomes 2 and 17, respectively, and showed associations with affected liver surface and IgM production. QTL5 and QTL6 linked to chromosomes 6 and 8, respectively, were related to worm recovery, eggs in tissues, and IgG2a.

Next, we examined the syntenic regions of the identified QTLs using Ensembl to explore the homology between mice and humans. For this study, we chose the QTL1 related to liver damage located on chromosome 15 with the marker peak SNP rs6169611 and QTL2 located on chromosome 18 with the marker peak SNP rs13483259. We found syntenic regions with QTL1, within the human genome on chromosomes 5, 8, 12, and 22, particularly with a homologous region on human chromosome 8, in the 8q24.3 region. QTL2 in mice had syntenic regions within the human genome on chromosomes 2, 5, 10, and 18, mainly on human chromosome 18, region 18q11.2-12.3. Previous studies on humans have also shown an association with the schistosomiasis susceptibility of specific chromosomal regions located on chromosomes 1, 5, and 6 [14]. Not all the regions studied were associated with susceptibility to infection by *S. mansoni*, except SM2 locus (6q22-23) associated with liver fibrosis. This suggests that the regions identified in our experimental backcross model, which exhibit homology in the human genome, have not been described in previous studies.

Finally, we analyzed the QTL association with the four groups with different levels of severity of schistosomiasis. We found that Group 4 (the most severe one) had a high percentage of heterozygous mice for most SNPs that showed linkage. However, the pathophenotypes we studied are complex and are influenced by intermediate phenotypes operating at different levels of biological organization. For this reason, the results obtained may not capture all the genetic factors that contributed to variability in the pathophenotypes. The linkage analysis of each phenotype revealed only those regions that contributed to this value with sufficient force. However, genetic determinants underlying the variation in intermediate phenotypes may not have exhibited sufficient association with the pathophenotypes in this analysis, potentially leading to a loss of heritability [33]. Our study has limitations considering the number of mice included, the small size effect of involved chromosomal regions, and the statistical power of QTL detection for the studied variables. Nonetheless, our results are the first steps toward identifying genomic regions that control the phenotypic variation observed for *S. mansoni* infections. The dimension of significant regions will need additional fine mapping to propose reliable candidate genes influencing the defense response.

We can conclude that a BX cohort was generated through backcross, with significant heterogeneity, helpful in studying the genetic influence on the susceptibility to infection produced by *S. mansoni*. Our study revealed 19 chromosomic regions throughout the mouse genome. Chromosomes 15 and 18 had the most relevant linkage to granulomas and hepatic surface damage, and these regions showed homology to human chromosomes 8 and 18. These results contribute to identifying target regions that regulate the variation in the complex parasite susceptibility trait in schistosomiasis. Implementing these results in control and treatment strategies could benefit populations particularly susceptible to the disease through different factors such as preventive treatment and personalized education based on genetic susceptibility, tailored patient treatment, the development of vaccines targeting genetics and intensified efforts toward the control of environmental factors or genetic counseling.

## 4. Materials and Methods

### 4.1. Animals and Parasites

A backcross (BX) mouse cohort was generated after two stages. Firstly, we crossed a mouse susceptible to schistosomiasis strain (CBA/2J JAX) with a resistant one (C57BL/6J JAX) to produce the F1B6CBA mice. Subsequently, the BX mice were generated by backcrossing F1B6CBA female mice with CBA/2J males. The parental strains were purchased from Charles River Laboratories (Lyon, France) [34]. The animal procedures complied with the regulations on animal experimentation (L 32/2007, L 6/2013 RD 53/2013, and 2010/63/CE). The Ethics Committee of the University of Salamanca reviewed and approved the study (Protocols Numbers 15/0018 and 12/3351). All mice were housed in the Animal Research Facility of the University of Salamanca under standard conditions with free access to water and food ad libitum. The welfare and health of the animals were monitored during the experiment following the guidelines of the Federation of European Animal Science Laboratory Associations (FELASA). The *S. mansoni* LB strain was routinely maintained on *Biomphalaria glabrata* snails and passages in CD1 mice every 2–3 months. Seven-week-old mice weighing 20–25 g were infected with cercariae of *S. mansoni*. The number and viability of the cercariae were determined using an Olympus SXZ9 stereomicroscope (Tokyo, Japan) [35].

### 4.2. BX Mouse Cohort Infection and Parasitological and Pathological Traits Obtained

The BX mice were percutaneously infected with 150 ± 5 *S. mansoni* cercariae per mouse using the ring method. Nine weeks after the infection, the mice were euthanized using a lethal dose of pentobarbital (150 mg/kg) intraperitoneal. Adult worms inside the portal and the mesenteric veins were recovered through perfusion with the help of an Olympus stereomicroscope SZX9 and a C-2000Z camera (Olympus Tokyo, Japan). Worms were obtained by opening the portal vein and extracting them with tweezers. The number of males and females were recorded. The liver and intestine were removed, weighed, and digested with 5% KOH overnight to determine the number of eggs in the tissues. Quantification was performed in triplicate with a McMaster chamber. The worm’s fecundity was calculated by dividing the number of eggs found in the liver and small intestine by the number of female worms [35]. Three photographs of different parts of each mouse liver were collected before perfusion to assess the extension of tissue lesions. The number of granulomas per square centimeter was counted by two qualified researchers separately in each of the three micrographs. Adobe ImageJ 1.45 software (NIH, Bethesda, MD, USA) was used to enhance the image contrast [36], and PowerPoint was used to set the sectors that facilitated the count. In case of disagreement, the intervention of a third investigator was requested to resolve the differences.

### 4.3. Peripheral and Spleen White Blood Cell Subpopulations Quantified via Flow Cytometry

Blood samples were mixed with heparin, and splenocytes were obtained with the aseptic perfusion of the spleen with 10 mL of sterile phosphate-buffered solution (PBS) at 37 °C. Samples of a minimum of 10^5^ peripheral white blood cells or splenocytes were mixed with 150 µL of lysis solution (154 mM NH_4_Cl, 10 mM KHCO_3_, and 0.082 mM EDTA (Sigma)) for 30 min and then centrifuged at 237× *g.* This process was repeated, and the pellet was resuspended in 150 μL of PBS containing 2% fetal bovine serum (PBS-FBS). Monoclonal antibodies conjugated with fluorescein isothiocyanate (FITC) against CD4, with phycoerythrin (PE) against CD8, with allophycocyanin (APC) against B220, and with peridinin–chlorophyll proteins (PerCP-Cy™ 5.5) against CD45 from BD Pharmingen (New Jersey, USA) were diluted at 1/50 in 25 μL of PBS, 2% FBS, per well and incubated for 30 min at room temperature in the dark. Then, cells were washed with 100 μL of PBS-FBS and resuspended in 200 μL of PBS-FBS at 4 °C in the dark. A FACSCalibur flow cytometer (BD Biosciences, Franklin Lakes, NJ, USA)) was used to collect a minimum of 30,000 events from each sample [37]. The data were analyzed using the freely available software Flowing Software 2.5.1 (http://flowingsoftware.btk.fi, Cell Imaging Core, Turku Centre for Biotechnology, accessed on 10 October 2020) [38].

### 4.4. Detection of Anti-S. mansoni Antibodies in Mice Infected with the Parasite Using ELISA

Serum samples were obtained both prior to infection and at necropsy to detect specific IgG, IgG1, IgG2a, and IgM. Polystyrene Costar 96-well plates (Costar^®^ 3596, Corning Inc., San Diego, CA, USA) were coated with 5 μg/mL of *S. mansoni* somatic antigen (SoSmAg) in carbonate buffer at pH 9.6 (100 μL/well) and incubated at 4 °C for 16 h. Subsequently, the plates were washed three times with PBS with 0.05% Tween 20 (PBST) (200 μL/well). Plates were blocked with bovine serum albumin (B4287, Sigma, Saint Louis, USA) 2% in PBST for 50 min at 37 °C (100 μL/well), and then they were washed three times with PBST. The sera were diluted 1:100 in PBST (100 μL/well), incubated for 1 h at 37 C in duplicate, and washed three times with PBST as above. Horseradish peroxidase-conjugated anti-mouse IgGI, IgG1, IgG2a-HRP, and IgM (Sigma) were used at 1:1000 in PBST (100 μL/well), incubated 1 h at 37 °C, and washed as above. Finally, the plates were developed using H_2_O_2_ (0.012%) and orthophenylenediamine (0.04%) in 0.1 M citrate buffer, pH 5.0 (100 μL/well). The reaction was stopped by adding 3N H_2_SO_4_ (50 μL/well) and read at 492 nm using a MultiSkan GO ELISA plate reader (Thermo Fisher Scientific, Vantaa, Finland) [35].

### 4.5. SNP Genotyping, Quality Control, and Linkage Analysis

DNA was extracted from mouse tails using the DNeasy^®^ Blood and Tissue Kit (Qiagen^®^, Hilden, Germany), and DNA concentration was determined with NanoDrop ND-1000 (Thermo Fisher Scientific Inc., Waltham, MA, USA). Genotyping was performed using Illumina’s Mouse Low-Density Linkage Panel Assay at the National Genotyping Center (CeGen) of the National Center for Oncological Research (CNIO) [32]. A platform with 961 informative SNPs between C57BL/6J and CBA/2J strains was used. We prepared a matrix with 20 phenotypic variables including sex and genotype data. In the matrix, autosomal female X homozygotes were coded as 0, heterozygotes as 1, and male X chromosome as 2. Data quality control was carried out using the *R*/*qtl* package in R3.3.3, verifying the detection of the backcross, identifying genotyping errors, and quantifying crossovers to exclude mice presenting less than 5 or more than 20 overcrossings. Additionally, a comparison between the expected genetic map for BX mice and the estimated map was performed. Genetic linkage analysis was conducted using the pathophenotypes to assess parasitological and pathological variables, cellular populations, and immunoglobulins. The genetic distances, derived from recombination frequencies between markers within the BX cohort, were compared with the genetic distance recorded for markers in the reference database Mouse Genome Informatics [8] (http://cgd.jax.org/mousemapconverter, accessed on 10 October 2010). The maximum-likelihood mapping method was employed, using the hope-maximization (HM) algorithm. The Haldane function was used to calculate the conditional genotype, with a step size of 2.5 cm and a genotyping error of 0.001 [39]. We used the LOD score to calculate the statistical significance of the linkage of the QTLs found. Additionally, we used the criteria established by Lander and Kruglyak [40] to determine the significance level of the observed association between a QTL and the phenotype of interest. Therefore, LOD score values higher than 1.4 were considered indicative of suggestive linkage. The Ensembl bioinformatics tool (https://www.ensembl.org/index.html, accessed on 10 October 2010) was used to identify syntenic regions between mice and human beings.

### 4.6. Data Analysis

The mean and standard error of the mean were determined for each variable, and the normal distribution was checked using the Kolmogorov–Smirnov test. In the case of a normal distribution, differences between groups were examined using ANOVA followed by post hoc Tukey’s test or Student’s *t*-test. In the case of non-normal distribution, differences were examined using the Kruskal–Wallis test. The Pearson correlation coefficient (r) was used to explore the relationships between two variables, and the statistical significance was performed using Student’s *t*-test.

Multivariate models were generated to study similarities among mice and consider the potential influence of sex on worm recovery, eggs in the liver and small intestine, fecundity, granulomas, and the affected liver surface. All the variables were standardized to 0 mean, and standard deviation was set to 1 to avoid the magnitude differences of the variables. Cluster analysis (CA) dendrograms were used to visualize the distances between the cases, aiding in the identification of patterns [41]. Principal components analysis (PCA) condenses information from multiple variables, identifying groups and the influence of each variable [42]. Additionally, we applied the k-means clustering algorithm to generate groups or clusters according to the observed severity levels in mice. Further analyses were performed within the obtained groups, including median proportions for each variable. A significant difference was considered in all tests when the *p*-value for the rejection of the null hypothesis was less than 0.05. The results were analyzed using the SIMFIT statistical package Windows version 7.3.7 [43].

## Figures and Tables

**Figure 1 ijms-24-14768-f001:**
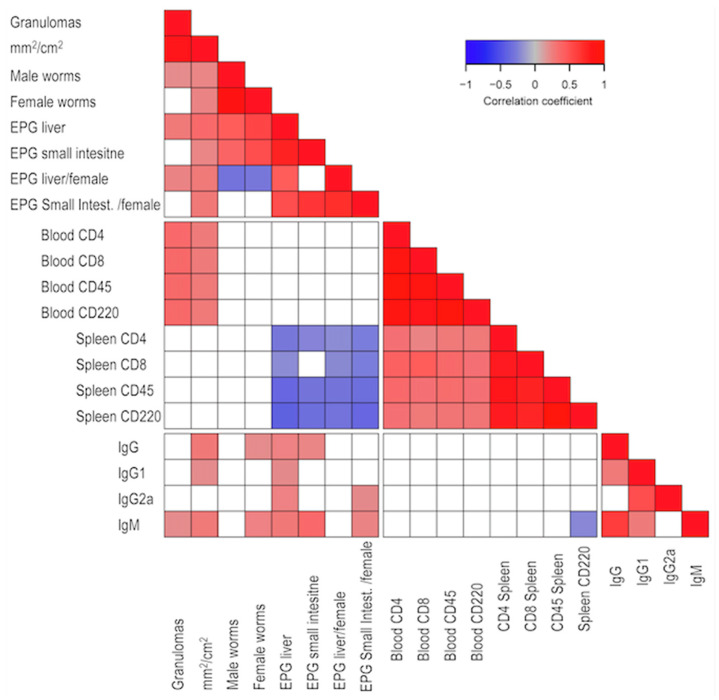
Correlation analysis between variables. Direct correlations are depicted in red, while inverse correlations are shown in blue. The intensity of the color reflects the correlation coefficient according to the scale. Only significant correlations (*p* < 0.05) are presented, while non-significant (*p* > 0.05) correlations are shown in white.

**Figure 2 ijms-24-14768-f002:**
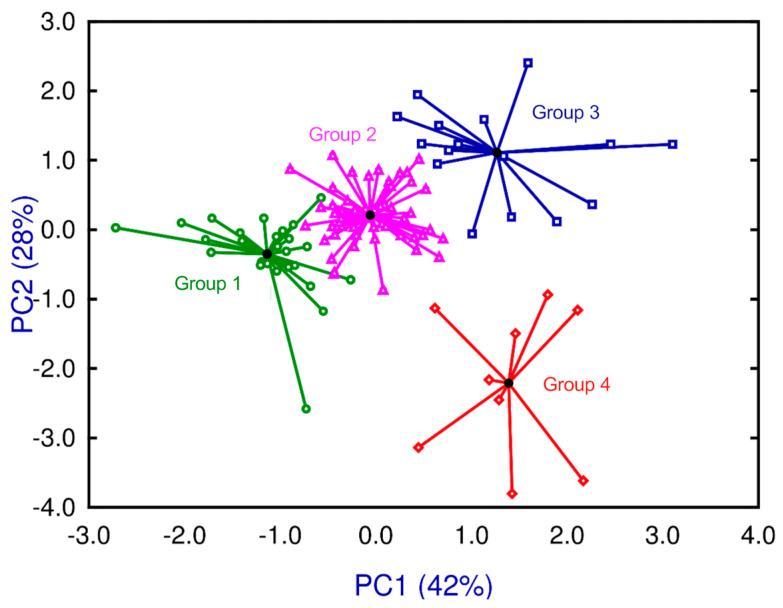
K-means cluster analysis shows four groups around their centroids and radios ranging in severity in four levels from mild (Group 1 in green), mild-moderate (Group 2 in pink), moderate-severe (Group 3 in blue), to severe (Group 4 in red).

**Figure 3 ijms-24-14768-f003:**
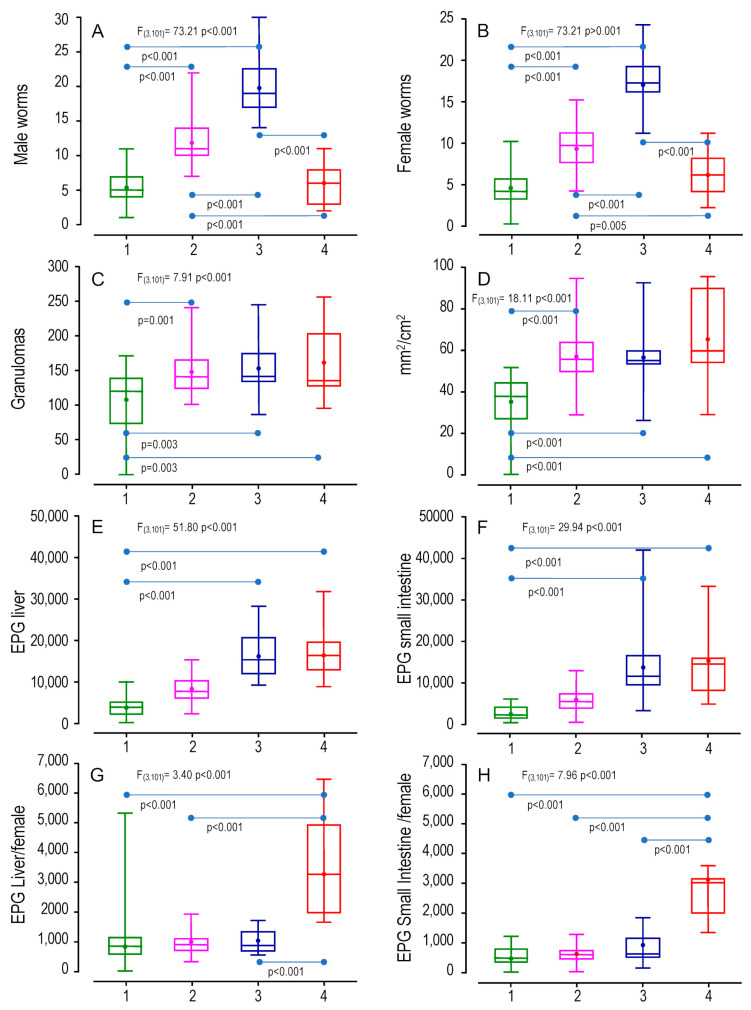
Differences in pathophenotypes and intermediate phenotypes among the four identified groups through k-means analysis. We evaluated the following variables: (**A**) male worms, (**B**) female worms, (**C**) liver granulomas, (**D**) affected surface of the liver (mm^2^/cm^2^), (**E**) eggs per gram (EPG) of liver granulomas, (**F**) EPG of the small intestine, (**G**) fecundity of females in the liver (EPG Liver/female), and (**H**) fecundity of females in the small intestine (EPG Small intestine/female). Differences among groups were studied using ANOVA, and pairwise comparison groups were assessed using the Tukey test.

**Figure 4 ijms-24-14768-f004:**
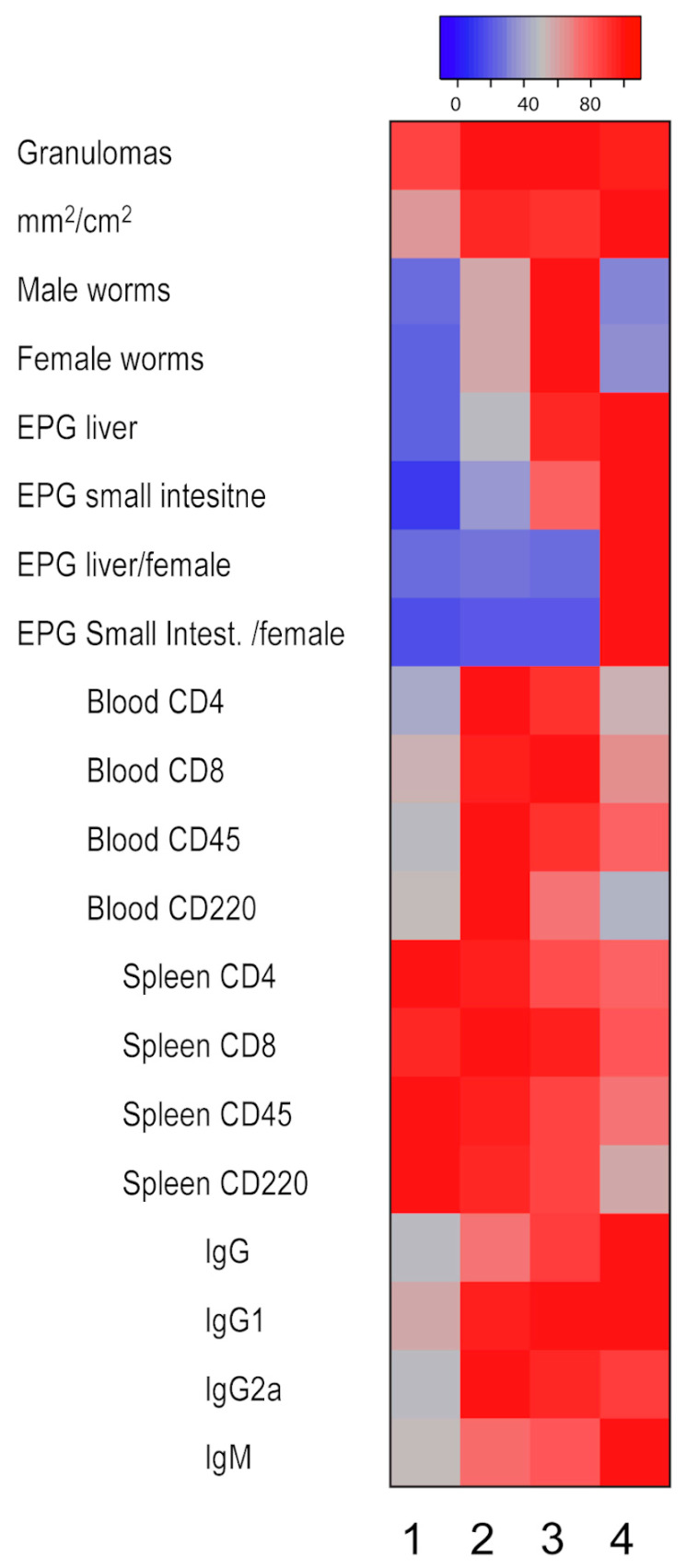
Analysis of the medians of the variables across the four groups of mice. The reference value was determined as the maximum median of each variable within the four groups. The proportion, represented as a percentage, indicates the variation experienced across the groups based on each group’s median in relation to the maximum. In terms of color representation, red indicates a high difference among median, while blue color signifies a low difference among medians.

**Figure 5 ijms-24-14768-f005:**
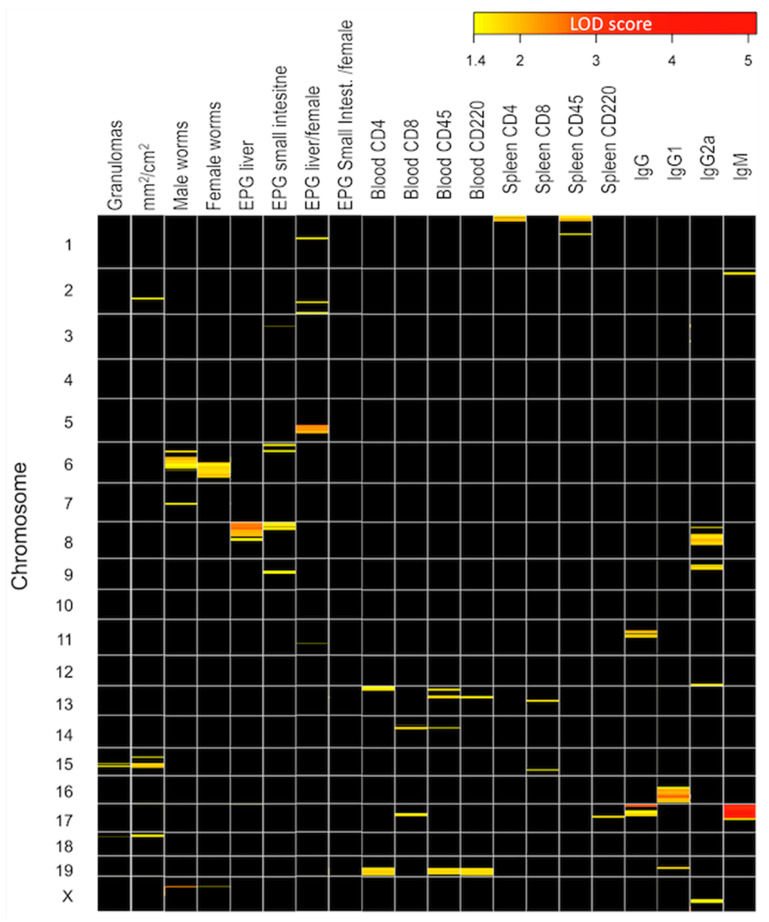
Heatmap of the chromosome QTL regions linked to pathophenotypes and intermediate phenotypes. The LOD score on an intensity scale from yellow to red represents the strength of the linkage. QTLs with LOD scores below 1.4 are represented in black.

**Figure 6 ijms-24-14768-f006:**
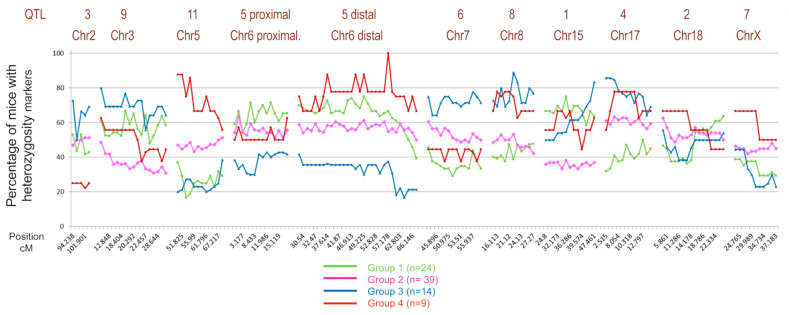
Comparison of the groups defined at the genetic level. The figure shows the percentage of mice with a particular heterozygous genetic marker, differentially presented in the four groups of mice according to the severity of the disease.

## Data Availability

The datasets generated and/or analyzed during the current study are available in the DIGITAL.CSIC repository, https://doi.org/10.20350/digitalCSIC/15081 (accessed on 20 September 2023).

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
