# Peer review of "Identification of Genomic Regions Implicated in Susceptibility to Schistosoma mansoni Infection in a Murine Backcross Genetic Model"

_ijms, 2023, doi:10.3390/ijms241914768_

Round 1

Reviewer 1 Report

The manuscript is concerned with studying genetic markers related to the severity of schistosomal infection, in terms of intensity of infection and its organic repercussions.

The authors used an experimental model that proved to be quite adequate for its purpose.

Also, the study takes up a very interesting theme, started at the end of the last century with studies conducted by Dessein and collaborators, studying human populations; the work, using an experimental model, indicates the presence of a greater number of genetic markers involved in the pathogenesis of schistosomiasis mansoni.

The methodology employed seems adequate and the text is well structured.

I have two observations to the authors:

1. line 297, pg10 "...but we don't observe any bias....". How do the authors explain this finding?

2. line 352, page 11: Could the authors exemplify ways on how to use their findings in control and treatment strategies to benefit populations particularly susceptible to this disease?

Author Response

Reviewer 1

Comment

The manuscript is concerned with studying genetic markers related to the severity of schistosomal infection, in terms of intensity of infection and its organic repercussions.

The authors used an experimental model that proved to be quite adequate for its purpose.

Also, the study takes up a very interesting theme, started at the end of the last century with studies conducted by Dessein and collaborators, studying human populations; the work, using an experimental model, indicates the presence of a greater number of genetic markers involved in the pathogenesis of schistosomiasis mansoni.

The methodology employed seems adequate and the text is well structured.

Answer:

We thank the reviewer apprecitations

I have two observations to the authors:

  1. line 297, pg10 "...but we don't observe any bias....". How do the authors explain this finding?

Answer:

We have observed significant changes in parasitological variables, in CD45 and IgG antibodies within the four detected groups but there are no significant differences associated with Th1 and Th2 response. To clarify we have changed “…but we don't observe any bias...." to …“but we don't observe any differences...." (L315 Tracked new version TNV)

  1. line 352, page 11: Could the authors exemplify ways on how to use their findings in control and treatment strategies to benefit populations particularly susceptible to this disease?

Answer

Regarding the schistosomiasis-susceptible population in a specific region, such as a rural community in Africa where a high prevalence of schistosomiasis is observed, genetic findings can make a difference. Imagine setting up temporary clinics that offer genetic testing to residents, identifying those most susceptible. People with a highly susceptible genetic profile would receive preventive treatment and personalized education on how to reduce exposure to the parasite. This personalized education, based on genetic susceptibility, could emphasize personal risks, and by being informed, individuals would be more motivated to take preventive measures, such as avoiding infested waters.

In a clinical context, consider a hospital in Brazil where a patient is diagnosed with schistosomiasis. With the knowledge gained about the chromosomal regions associated with increased susceptibility or response to infection, the medical team could tailor the patient's treatment, opting for drugs or doses that best suit their genetic profile.

As research progresses, developing vaccines targeting genetics could also be possible. In places like Egypt, if a subpopulation is found to have a particular genetic susceptibility, that community could be prioritized for administering a new vaccine tailored to their specific needs.

Beyond direct treatment and prevention, the environment also plays a crucial role. In regions with a high proportion of susceptible individuals, efforts could be intensified to control environmental factors, such as eradicating snails that act as intermediate hosts for the parasite or introducing predatory fish of these snails to reduce their population.

Looking ahead, couples wishing to have children could benefit from genetic counseling. If both parents carry genes associated with high susceptibility, they may receive recommendations on specific preventive measures to protect their children from an early age.

In summary, integrating genetic findings into schistosomiasis treatment and prevention strategies could offer more personalized and effective solutions for communities most affected by this disease.

We have added some exemples of the use of these findings (L373-375 TNV)

Reviewer 2 Report

The manuscript by Hernández-Goenaga et al. is well written and organized. This work provides the first report of some significant quantitative trait loci (QTL) related to the susceptibility to Schistosoma mansoni infection.

The results were clearly presented and discussed with accuracy. Therefore, I consider that the manuscript deserves publication.

A few minor modifications and concerns listed below.

Line 48: on SNPs?

Line 48: The term SNPs (single nucleotide polymorphisms) should be explained in this sentence because it is the first time it appears in the manuscript. Currently, the definition can be read in lines 72-73, so it is necessary to move it to line 48.

Line 51: There is a discordance in the sentence. Is lacking some information? After the comma is a capital letter.

Figure 2: I assume that “Cluster” and “Group” are referring to the same concept but, to prevent confusion, it is preferable to use the same word. For instance, the image shows “Cluster 1”, but caption says “Group 1”

Lines 202-207: I suggest moving this paragraph to section 4. Materials and Methods. Section 3.5 would start with “Through our analysis,…”

Lines 450-451: In the case of Non-normal distribution, the differences between groups were examined too?

Figure S1 captions. The descriptions of the parasitological variables are confusing. ¿B are female and pathological worms? A possible option is the following:

Representation of parasitological variables: (A) number of recovered male worms, (B) number of recovered female worms, (C) eggs per gram of liver and (D) number of granulomas/cm2 of liver surface between parental strains (CBA/2J, C57/2J) F1 progeny (F1B6BCA) and Backcross (F1BCX). The means and the standard error of the mean (SEM) were represented.

Figure S3 captions. Lines 518-519: Boxplot or Box plot? Also, there is an unnecessary parenthesis in “of)”.

Author Response

Dear reviewer

We have considered your comments and we have followed your suggestions and we have made the modifications in the ms.

We hope our answers are satisfactory.

Best regards. Julio and Antonio.

Reviewer 2

Comments

The manuscript by Hernández-Goenaga et al. is well written and organized. This work provides the first report of some significant quantitative trait loci (QTL) related to the susceptibility to Schistosoma mansoni infection.

The results were clearly presented and discussed with accuracy. Therefore, I consider that the manuscript deserves publication.

A few minor modifications and concerns listed below.

Answer:

We thank the reviewer appreciations.

Line 48: on SNPs?

Line 48: The term SNPs (single nucleotide polymorphisms) should be explained in this sentence because it is the first time it appears in the manuscript. Currently, the definition can be read in lines 72-73, so it is necessary to move it to line 48.

Answer:

We have incorporated the suggestion of the reviewer. We have deleted “(single nucleotide polymorphisms) on Lines 72-73 Tracked new version (TNV)  and we have incorporated in Line 48 TNV.

Line 51: There is a discordance in the sentence. Is lacking some information? After the comma is a capital letter.”

Answer:

We thank the reviewer and we have revised the sentence to make it sense.

Figure 2: I assume that “Cluster” and “Group” are referring to the same concept but, to prevent confusion, it is preferable to use the same word. For instance, the image shows “Cluster 1”, but caption says “Group 1”

Answer:

We have revised the use of “cluster” and “group” words. Cluster was maintained in methodology reference (K-means cluster analysis) and we changed to “Group” when we are talking aggrupation of mice: in Figure 2, Lines 154, 158, 161, 163, 349 TNV.

Lines 202-207: I suggest moving this paragraph to section 4. Materials and Methods. Section 3.5 would start with “Through our analysis,…”

Answer:

We have eliminated methodology information from lines 202-207 of the initial version. Only remain one sentence to say that three SNP and six mice were excluded after the quality control (Lines 210-211 TNV). Moreover, we have revised the 4.5 section to avoid repetitions (Lines 454-455 TNV).

Lines 450-451: In the case of Non-normal distribution, the differences between groups were examined too?

Answer:

We have used the non-parametric test (Kruskal-Wallis) to identify differences between groups. We have added a sentence in the new version in 4.6 section to indicate this forgotten technique (Line 475-476 TNV).

Figure S1 captions. The descriptions of the parasitological variables are confusing. ¿B are female and pathological worms? A possible option is the following:

Representation of parasitological variables: (A) number of recovered male worms, (B) number of recovered female worms, (C) eggs per gram of liver and (D) number of granulomas/cm2 of liver surface between parental strains (CBA/2J, C57/2J) F1 progeny (F1B6BCA) and Backcross (F1BCX). The means and the standard error of the mean (SEM) were represented.

Answer:

We thank your suggestion. We have modified the figure title of Figure S1 (Line 533-535 TNV)

Figure S3 captions. Lines 518-519: Boxplot or Box plot? Also, there is an unnecessary parenthesis in “of)”.

Answer:

We have revised the ms and we have selected “Box plot”, therefore we have modified the “Boxplot” to “Box plot in L545 TNV.

We have deleted the unnecessary parenthesis in L545 TNV.
